# The Issue of Pharmacokinetic-Driven Drug-Drug Interactions of Antibiotics: A Narrative Review

**DOI:** 10.3390/antibiotics11101410

**Published:** 2022-10-13

**Authors:** Dario Cattaneo, Cristina Gervasoni, Alberto Corona

**Affiliations:** 1Unit of Clinical Pharmacology, ASST Fatebenefratelli Sacco University Hospital, via GB Grassi 74, 20157 Milan, Italy; 2Gestione Ambulatoriale Politerapie (GAP) Outpatient Clinic, ASST Fatebenefratelli Sacco University Hospital, 20157 Milan, Italy; 3Department of Infectious Diseases, ASST Fatebenefratelli Sacco University Hospital, 20157 Milan, Italy; 4Accident & Emergency and Anaesthesia and Intensive Care Medicine Department, Esine and Edolo Hospitals, ASST Valcamonica, 25040 Brescia, Italy

**Keywords:** antibiotics, critically ill patients, drug-drug-interactions, pharmacokinetics

## Abstract

Patients in intensive care units (ICU) are at high risk to experience potential drug-drug interactions (pDDIs) because of the complexity of their drug regimens. Such pDDIs may be driven by pharmacokinetic or pharmacodynamic mechanisms with clinically relevant consequences in terms of treatment failure or development of drug-related adverse events. The aim of this paper is to review the pharmacokinetic-driven pDDIs involving antibiotics in ICU adult patients. A MEDLINE Pubmed search for articles published from January 2000 to June 2022 was completed matching the terms “drug-drug interactions” with “pharmacokinetics”, “antibiotics”, and “ICU” or “critically-ill patients”. Moreover, additional studies were identified from the reference list of retrieved articles. Some important pharmacokinetic pDDIs involving antibiotics as victims or perpetrators have been identified, although not specifically in the ICU settings. Remarkably, most of them relate to the older antibiotics whereas novel molecules seem to be associated with a low potential for pDDIs with the exceptions of oritavancin as potential perpetrator, and eravacicline that may be a victim of strong CYP3A inducers. Personalized therapeutic drug regimens by means of available web-based pDDI checkers, eventually combined with therapeutic drug monitoring, when available, have the potential to improve the response of ICU patients to antibiotic therapies.

## 1. Introduction 

Patients in intensive care units (ICU) differ considerably from those in other hospital wards having a higher level of sickness severity, requiring tailored and aggressive medical interventions, and often present with or contract severe infections. Multidrug-resistant organisms are also common in this setting [1]. As a result, these patients have substantial mortality rates (40–65%), particularly if they have a high severity of illness score, sepsis, and septic shock [2,3].

Given this background, immediate and appropriate antibiotic therapy is mandatory to improve the clinical outcome of ICU patients [4]. Antibiotic therapy for these patients is initially empirical but revised when the results of the microbiological tests become available. Usually, concomitant medications are not considered as a key factor in the selection of the antibiotic therapy because the potential drug-drug interactions (pDDIs) are not considered a clinical issue, given the relative short time of antibiotic treatments (usually less than 7–10 days). However, it must be considered that ICU patients are usually elderly, with multi-comorbidities and on heavy polypharmacy [5,6]. For these reasons, pDDIs are likely to take place in the ICU setting when antibiotic therapies are added on the top of an important background of maintenance medications, eventually leading to treatment failure or development of drug-related toxicity [7,8].

In this review, we firstly examine the main pharmacological concepts related to pDDIs with a focus on those that are driven by pharmacokinetics. The second section of the manuscript deals with antibiotics as both victims and perpetrators of pharmacokinetic pDDIs in ICU adult patients, with a focus on the most recently marketed molecules. Some suggestions on how to handle these pDDIs in the daily clinical practice are also given.

### 1.1. Search Strategy

This narrative review aimed to summarize what is known to date about the pDDIs involving antibiotics to treat recurrent infections mainly in ICU patients. The literature was selected through a search for relevant papers in the PubMed database published from January 2000 to June 2022 using the search terms.

“Drug-drug interactions”, “pharmacokinetics”, “antibiotics”, and “ICU” or “critically-ill patients”. Moreover, additional studies were identified from the reference list of retrieved articles. Only articles published in English were considered.

### 1.2. The Clinical Relevance of pDDIs

PDDIs represent a highly complex aspect of clinical pharmacology because, unfortunately, most of the information produced during the development of a drug is not very useful in determining their clinical relevance. Furthermore, most of what we know about the possible impact of pDDIs on humans comes from experimental models or studies of healthy volunteers in situations that are very different from the everyday clinical context, in which drugs are actually used once they are marketed [9]. Consequently, the information collected in pre-marketing studies should only be considered a starting point for a more comprehensive bedside approach that takes into account all of the other factors that largely govern the risk of pDDIs. This risk is directly proportional to the number of drugs received, but it is also necessary to remember that some patients (i.e., ICU elderly patients with excretory organ deficiency on renal replacement therapies, etc.) are more at risk than others to experience clinically relevant pDDIs [9].

### 1.3. The Mechanisms Underlying pDDIs

The most widely studied and clinically understood mechanism for pDDIs is pharmacokinetics (i.e., the capacity of a molecule to interfere with the absorption, distribution, metabolism, or elimination of another drug) [9]. pDDIs involving orally administered drugs—as the well-known chelating effect of bi- and trivalent cations (calcium, aluminum, magnesium, iron, etc.) resulting in reduced absorption of fluoroquinolones or tetracyclines in the general patient population—are insignificant in the ICU settings, because the patients are in most cases intubated or unconscious and, therefore, are unlikely to be treated with oral drugs.

The most frequent pharmacokinetic-driven pDDIs in ICU involve the inhibition or induction of drug metabolizing enzymes [7,8]. The most common are those affecting drug metabolism due to induction or inhibition of the cytochrome P450 (CYP) leading to abnormal drug exposure; among the different CYP isoforms, CYP3A, CYP2D6, and CYP2B6 are those most frequently involved in DDIs [9]. Over the last few years, considerable attention has also been given to DDIs involving the transmembrane proteins (such as P-glycoprotein, breast cancer resistance protein, etc.) that act as carriers of various drugs [10,11]: for example, the trimethoprim-induced inhibition of organic cation transporter 2 (OCT2) can significantly increase the bioavailability of metformin by blocking its renal tubular elimination [12]. Conversely, pharmacodynamic-based pDDIs may involve the combined (synergistic, agonistic, or antagonistic) effects of two or more molecules on the same pharmacological target, such as addictive/synergistic effects of fluoroquinolones or macrolides with co-medications affecting the QT prolongation, or different targets, such as the combined use of a beta lactam and a beta lactamase inhibitor) [13]. Here, we will focus only on pharmacokinetic-driven pDDIs.

### 1.4. Pharmacokinetic Issues in ICU

The achievement of optimal antimicrobial exposure is difficult in clinical practice because most of these drugs are administered according to standard dosing regimens which do not take into account pathophysiological and/or iatrogenic factors that are likely to affect the pharmacokinetics in ICU patients. This makes the management of antimicrobial therapy in these patients extremely challenging. The effects of altered pathophysiology in ICU patients on the pharmacokinetics of antimicrobial agents have been recently reviewed [14] and are briefly summarized below.

The most frequently altered pharmacokinetic parameter in ICU patients is the volume of distribution. Infections result in a significant increase in the production of endogenous mediators which can increase capillary permeability resulting in a shift of the fluids from the intravascular compartment to the interstitial space. These events cause a significant “dilution” of the systemic concentrations of antibiotics characterized by a low volume of distribution (i.e., less than 20 L such as beta-lactams, aminoglycosides), resulting in suboptimal drug exposure [15].

Approximately 35–40% of ICU patients are severely hypoalbuminemic (serum albumin concentrations <2 g/dL) [16]. This needs to be taken into account when these patients are treated with highly bound antimicrobial agents (those with protein binding >80%), especially if these drugs have some degree of renal elimination. The reduced concentration of albumin is likely to increase the free drug fraction available for elimination through the kidneys resulting in sub-therapeutic drug concentrations [16]. 

Another important clinical condition that needs to be carefully considered in ICU is altered renal function, especially when hydrophilic antimicrobials are administered. Acute kidney injury results in reduced drug excretion whilst augmented renal clearance is associated with a 2-to-8-fold increase in the clearance of renally excreted drugs, such as the beta-lactam antibiotics [17].

Approximately 5% of ICU patients are treated with continuous renal replacement therapies instead of intermittent hemodialysis to better maintain hemodynamic stability [18]. However, pharmacokinetic data of many antimicrobial agents in continuous renal replacement therapy is largely lacking. Finally, drugs metabolized by the hepatic route may be affected in ICU patients with acute or chronic forms of hepatic dysfunction caused by infection associated with hepatocellular injury, ischemia, hemolysis, or direct damage from drug-related hepatotoxicity [14].

The acknowledgement of the pharmacokinetic alterations In the ICU patients is important for a better understanding of the pDDIs that are likely to take place in this clinical setting. For instance, it is well known that concomitant administration of probenecid may increase the systemic exposure of beta-lactams [19]. However, as these antibiotics are characterized by a very low volume of distribution and a renal elimination, concomitant administration of probenecid and a beta-lactam in an ICU patient with hypoalbuminemia experiencing augmented renal clearance and fluid gain may apparently result in a reduction in the systemic exposure of the beta-lactam because the pathological conditions associated with ICU outweigh the effect of the DDI related to concomitant probenecid administration (Figure 1).

## 2. pDDIs Issues in ICU

Patients admitted to the ICU are at high risk for pDDIs due to the significant number of drugs prescribed and the complexity of drug regimens in this clinical setting [7,8]. As a consequence, an analysis of risk factors for adverse events in ICU patients reported that an increasing number of medications and DDIs was associated with a higher risk of injury [20].

According to available literature with regard to DDIs specifically involving anti-infective agents, a significant number of patients experienced at least one DDI during their ICU admission, with the antifungal drug fluconazole ranking in the top-ten DDIs, followed by aminoglycosides and macrolides [7,8]. More recently, Kusku and co-workers [21] analyzed data from 5 different hospitals and reported that DDIs with antimicrobial agents represented 26% of all interactions, with 42% and 38% of them “contraindicated” and “major”, respectively according to the Micromedex online reference system. Notably, apart from the azoles, quinolones, metronidazole, linezolid, and clarithromycin were responsible for 92% of the reported DDIs. In multivariate analysis, the number of prescribed antimicrobial agents (odds ratio: 2.3), prescribed drugs (odds ratio: 1.2), and hospitalizations in a university hospital (odds ratio: 1.8) were independent risk factors for developing DDIs. Similarly, Mehralian and co-workers in their cross-sectional prospective study found that 60% of ICU patients had at least one DDI [22]; nearly 87% of them, involving mainly antibiotics, were scored as harmful. Of particular relevance, DDIs involving metronidazole, azoles, azithromycin, and quinolones have been associated with QT prolongation [7,23]. 

It is clear that the implementation of appropriate programs and interventions aimed to reduce the frequency of DDIs of antibiotics in ICU is critical.

## 3. Antibiotics as Victims or Perpetrators of pDDIs

In the next chapters, we describe the main pDDIs involving the main classes of antibiotics with a focus on novel molecules recently marketed (summarized in Table 1).

### 3.1. Beta-Lactams

#### 3.1.1. Penicillins

Overall, penicillins are characterized by a low risk of pDDIs. As perpetrators, penicillins may reduce the disposition of oral contraceptives [24], however, this pDDI is unlikely to be of clinical relevance in the ICU setting. Penicillins may also reduce the renal excretion of methotrexate resulting in increased systemic exposure, whereas concomitant administration of probenecid or non-steroidal anti-inflammatory agents may increase the exposure of penicillins [25,26,27]. Also in these cases, the clinical relevance of these pDDIs in ICU may be questionable.

One potential exception may be represented by flucloxacillin, a narrow-spectrum antibiotic of the group of isoxazolyl penicillins (semi-synthetic derivative). Indeed, different studies and case reports have consistently shown that concomitant administration of flucloxacillin can lead to a clinically-relevant reduction in the systemic exposure of antifungal azoles (voriconazole, posaconazole, and isavuconazole) (Table 2) [28,29,30,31,32,33]. Presumably the underlying mechanism is activation of the pregnane X receptor by flucloxacillin, which can induce CYP enzymes, uridine glucuronosyl transferase and/or P-glycoprotein [30]. Caution should be taken, therefore, when combining flucloxacillin and triazoles, because interactions may lead to suboptimal treatment of invasive fungal infections.

**Table 2 antibiotics-11-01410-t002:** Studies that have documented drug-drug interactions between flucloxacillin and antifungal azoles.

Study	Study Design	Flucloxacillin Dose	Azole/Dose	Main Findings
[28]	Case report	8 g/day	VRC 400–1000 mg/day	VRC trough fell to <1 mg/L and remained subtherapeutic until flucloxacillin discontinuation
[29]	Retrospective, 20 patients	1–12 g/day	Not reported	11/20 patients had VRC trough <1 mg/L (median 0.2 mg/L)
[30]	Case report 1	12 g/day	VRC, 4–8 mg/kg bid	1st VRC trough: <0.2 mg/L; 2nd VRC trough: <0.2 mg/L; 3rd VRC trough: 3 mg/L (after flucloxacillin discontinuation)
[30]	Case report 2	12 g/day	ISA 200 mg/day	ISA trough increased from <0.3 mg/L to 1.7–5.2 mg/L after flucloxacillin discontinuation
[31]	Case report 1	8 g/day	VRC 4–8 mg/kg bid; ISA 200 mg/day	VRC trough: 0.6 mg/L; ISA trough increased from 0.4 to 2 mg/L after flucloxacillin discontinuation
[31]	Case report 2	12 g/day	VRC 300 mg bid ISA 200 mg/day/bid	VRC trough: <0.2 mg/L; ISA trough increased from 0.6–1.5 mg/L to 2.6–5.1 mg/L after flucloxacillin discontinuation
[32]	Retrospective, 33 patients	Not reported	Not reported	VRC trough: 0.5 (0–1.8) mg versus 3.5 (1.7–5.1) mg/L in patients given or not flucloxacillin
[33]	Case report	8 g/day	VRC 200 mg bid; POS 300 mg bid	VRC trough reduced from 2.2 to <0.2 mg/L after adding g; POS trough reduced from 1.4 to 0.8 mg/L after adding flucloxacillin

VRC: voriconazole; ISA: isavuconazole; POS: posaconazole.

#### 3.1.2. Cephalosporins

The potential of cephalosporins to cause or to be victims of pDDIs is very similar to what has been already described for penicillins;many molecules of this class are unlikely to cause DDIs. Conversely, the bioavailability of most molecules of this class can be significantly increased by concomitant probenecid administration [25,26,27].

Ceftobiprole is a fifth-generation cephalosporin with a broad spectrum of activity against gram-negative pathogens approved for the treatment of hospital-acquired pneumonia (excluding ventilator-associated pneumonia) and community-acquired pneumonia [34]. Ceftobiprole is an inhibitor of the hepatocyte uptake transporters, organic anion transporting polypeptides (OATP) 1B1, and OATP1B3 [34]. These polypeptides act as uptake transporters, specifically expressed in the liver, for many drugs, including statins. Their inhibition (mediated by cefiderocol) might potentially result in reduced drug metabolism and increased systemic exposure [35]. However, such DDIs have never been reported in scientific literature.

Ceftaroline fosamil is the prodrug of ceftaroline, a fifth-generation parental oxyimino cephalosporin with bactericidal activity against *methicillin-resistant Staphylococcus aureus* [36]. The interaction potential of ceftaroline on medicinal products metabolised by CYP enzymes is expected to be low since it is neither an inhibitor nor an inducer of CYP enzymes. Ceftaroline is not metabolized by CYP enzymes, therefore co-administered CYP inducers or inhibitors are unlikely to influence the pharmacokinetics of ceftaroline. Ceftaroline is neither a substrate nor an inhibitor of renal uptake transporters [OCT2, organic anion transporter (OAT) 1 and OAT3] in vitro [36]. Therefore, interactions of ceftaroline with medicinal products that are substrates or inhibitors (e.g., probenecid) of these transporters would not be expected. 

Cefiderocol is a novel catechol-substituted cephalosporin antibiotic able to enter the bacterial periplasmic space as a result of its siderophore-like property [37]. Initial in vitro experiments indicated a potential inhibition of OAT1, OAT3, OCT1, OCT2 multidrug and toxin extrusion protein 2K (MATE-2K), and OATP1B3; however, a clinical trial in healthy volunteers concomitantly receiving cefiderocol with probe substrates indicated that cefiderocol had no clinically relevant impact on the pharmacokinetic of the probe substrates (reviewed in [36]). Similarly, based on data from in vitro experiments and phase I trials, no clinically relevant pDDIs are expected, and this is the same for ceftolozane, the other last generation cephalosporin marketed [38].

#### 3.1.3. Monobactams

Concomitant administration of furosemide or probenecid can slightly increase the systemic exposure of aztreonam [25,26,27]. The clinical relevance of this pDDIs is very limited.

#### 3.1.4. Carbapenems

As for the other beta-lactams, concomitant administration of probenecid can increase the systemic exposure of carbapenems [25,26,27]. Unlike other beta-lactams, concomitant administration of carbapenems can significantly reduce the systemic exposure of the antiepileptic drug valproic acid (reviewed in [39]). Valproic acid serum concentration generally returns to normal within two weeks after discontinuation of the carbapenem antibiotic. This effect, described for the first time nearly 20 years ago, has been consistently reported for ertapenem, imipenem, doripenem, and meropenem, with the latter associated with the highest reduction of valproate serum concentrations (exceeding, in some instances, 100%) [39]. Remarkably, episodes of seizures associated with concomitant valproate-carbapenem administration have been consistently reported, making this DDI of high clinical relevance [40,41,42,43]. The exact mechanism of the interaction is not known yet. The rapid onset of the DDI (within 24 h) and the 7–14 day recovery period after stopping carbapenem suggest that the mechanism of this DDI could be based on enzyme inhibition. Accordingly, it has been proposed that carbapenems may decrease intestinal absorption of valproate (which does not explain the interaction as it is also seen with valproate administered intravenously), inhibit valproate glucuronide hydrolysis, induce valproate hepatic glucuronidation, increase the renal clearance of valproate glucuronide and/or increase the distribution of valproate into red blood cells (reviewed in [38]). The valproate-carbapenem coadministration represents an optimal example on how DDIs, if properly understood, could be used to manage some clinical conditions. Indeed, extensive evidence is now available, showing that carbapenems can be successfully used to treat cases of valproate intoxication [44,45].

In 2017, Mahmoudi and co-workers reported the case of a critically ill patient which repeatedly required profound voriconazole dose reduction when high-dose meropenem was added [46]. The authors also performed in vitro assessments providing for the first time evidence that meropenem may inhibit the CYP2C19 and CYP3A4 isoforms. They concluded that during meropenem treatment, narrow therapeutic index drugs metabolized by these cytochromial enzymes require close monitoring and, eventually, dose reductions. It must be considered, however, at the moment, this is the only published case report showing a potential inhibitory effect of meropenem on phase I metabolic enzyme. Therefore, the clinical value of this pDDI remains to be established.

### 3.2. Glycopeptides and Lipoglycopeptides

Vancomycin and teicoplanin are not expected to act as victims or perpetrators of pharmacokinetic-based pDDIs [47,48].

Dalbavancin and oritavancin are new lipoglycopeptides characterized by a very long half-life (>200 h) recently approved for the treatment of acute bacterial skin and skin structure infections. In nonclinical studies and a population pharmacokinetic analysis, coadministration of dalbavancin with known CYP substrates, inhibitors, and inducers did not have a clinically significant effect on its pharmacokinetics (reviewed in [48,49]). Therefore, there is minimal potential for dalbavancin to cause clinically relevant DDIs. The same applies also for telavancin (another glycopeptide characterized, however, by a short half-life), which is not an inhibitor, inducer, or a substrate for CYP isoenzymes [50]. It is not yet clear whether dalbavancin or telavancin are substrates for hepatic absorption and efflux transporters.

Oritavancin has been studied in healthy volunteers to evaluate the concomitant use of the 1200-mg dose with probe substrates for several CYP enzymes (reviewed in [49]). In these studies, oritavancin was found to be a nonspecific, weak inhibitor of CYP2C9 and CYP2C19, and an inducer of CYP3A4 and CYP2D6, identified based on interactions with known substrates of these enzymes. Coadministration of oritavancin resulted in a 31% increase in the mean AUC of warfarin (CYP2C9), a 15% increase in the ratio of omeprazole to 5-hydroxy-omeprazole (CYP2C19), an 18% decrease in the mean area under the curve (AUC) of midazolam (CYP3A4), and a 31% decrease in the ratio of dextromethorphan to dextrorphan concentrations (CYP2D6) in the urine. Specifically, clinicians should be aware that coadministration of oritavancin with warfarin may result in higher exposure to warfarin, increasing the risk of bleeding.

### 3.3. Tetracyclines

The tetracyclines are well-known for DDIs involving chelation and reduced absorption of the antibiotic [10]. Drug products containing iron, magnesium, aluminum, or calcium reduce, to varying degrees, the bioavailability of all tetracyclines. Other drugs known to reduce the bioavailability of tetracyclines include bismuth subsalicylate, cholestyramine, and colestipol. As perpetrators, tetracyclines may reduce the disposition of oral contraceptives [51]. However, both these pDDIs are unlikely to be of clinical relevance in ICUs because of the limited use of oral drugs in this clinical context.

Tigecycline is the first glycylcycline to be launched and the first new tetracycline analogue marketed since minocycline over 40 years ago. In vitro experiments with liver microsomes confirmed that there is little potential for pDDIs for tigecycline, (reviewed in [52]). 

Eravacycline is a novel fluorocycline of the tetracycline class of antimicrobial agents that has activity against a broad spectrum of bacterial pathogens, including multidrug-resistant gram-positive and gram-negative organisms. Several studies in healthy humans have investigated the effect of a CYP3A4 inhibitor and inducer on eravacycline pharmacokinetics (reviewed in [53]). In one DDI study, a reduction in total eravacycline exposure of approximately one third and an increase in clearance of approximately 50% occurred with concomitant rifampin administration. The same group found that mean area under the concentration curve from time zero to the last quantifiable concentration (AUC0–t) and half-life was increased approximately 30–40% after a concomitant dose of eravacycline and itraconazole, and clearance was subsequently decreased. Taken together, these data indicate that the dose of eravacycline should be increased when given with a potent inducer of CYP3A, such as rifampin, but it is not clear that a dose adjustment should be made with an inhibitor such as itraconazole [53].

Omadacycline is a derivative of minocycline and a novel, first-in-class, aminomethylcycline antibiotic. Although no studies have been reported about pDDIs and absorption of omadacycline, it is advisable to avoid the concurrent administration of divalent- or trivalent cation-containing products before and for at least 4 h after oral administration of omadacycline (reviewed in [54]). Intravenous (IV) solutions containing multivalent cations (e.g., magnesium or calcium) should not be administered through the same IV line as omadacycline. In vitro studies with human liver microsomes have documented that omadacycline at clinically relevant concentrations have little to no reversible inhibition or induction of CYP isoforms, phase II metabolic enzymes and/or on the main drug transporter [54].

### 3.4. Macrolides

Macrolides continue to be an important therapeutic class of drugs with established efficacy in a variety of skin infections. Erythromycin, the prototype of macrolide antibacterials, is associated with a number of drawbacks, including a narrow spectrum of activity, unfavorable pharmacokinetic properties, and a significant number of pDDIs related to the inhibition of the CYP enzymes (reviewed in [55]). The development and marketing of newer macrolides improved the drug interaction profile associated with this class. Clarithromycin has lower affinity with liver enzymes, hence less involvement in drug interactions. However, carbamazepine and cyclosporin require a close monitoring when used with clarithromycin. The most success in avoiding drug interactions related to the inhibition of CYP has been through the development of the azalide subclass, of which azithromycin is the first and only to be marketed. Azithromycin has not been demonstrated to inhibit the CYP system in studies using a human liver microsome model, and to date has produced none of the classic drug interactions characteristic of the macrolides [56].

### 3.5. Fluoroquinolones

#### 3.5.1. Fluoroquinolones as Victims

Several studies focused on interactions between di- and trivalent metallic agents and fluoroquinolones (reviewed in [57]). Oral drug preparations (including gastric acid-reducing agents, multivitamins and OTCs) that contain multivalent cations are well known to chelate with fluoroquinolones in the gastrointestinal tract; co-administration may lead to clinically significant decreases (ranging from 30 to 70%) in oral fluoroquinolone bioavailability and an overall increase in fluoroquinolone-resistant bacteria. However, this DDI, as already pointed out, is of limited relevance in the ICU settings. Concomitant treatment with sevelamer hydrochloride, a phosphate-binding polymer, with ciprofloxacin should be avoided as it reduced the drug AUC by 50% [58].

#### 3.5.2. Fluoroquinolones as Perpetrators

Several fluoroquinolones appear to inhibit CYP1A2, albeit to different extents. Drugs undergoing biotransformation through CYP1A are, therefore, at risk of interacting with fluoroquinolones (reviewed in [59]). Caffeine drug exposure has been increased by ciprofloxacin with an AUC increase ranging from 20 to 145%, depending on the ciprofloxacin dose (200–1500 mg). Ciprofloxacin also increases clozapine and its metabolite N-desmethylclozapine serum concentration through CYP1A2 inhibition by approximately 30%. Several case reports suggest ciprofloxacin exhibits a drug interaction with olanzapine, probably through CYP1A2 inhibition, resulting in QT prolongation [59].

Theophylline is also a CYP1A2 substrate. Therefore, patients receiving this agent are at risk for a pDDIs of theophylline with several fluoroquinolones. Ciprofloxacin, in a dose of 1000 mg, reduced theophylline clearance by 20–30% [9]. 

Ciprofloxacin also appeared to exhibit an interaction with anesthetics ropivacaine and lidocaine by means of CYP1A2 inhibition, resulting in an increased anesthetic exposure (around 25–35%). Ciprofloxacin was found to greatly increase the AUC and maximum concentration (Cmax) of tizanide, a centrally acting muscle relaxant that is metabolized mainly by CYP1A2, 874%, and 583% respectively [59]. Because of escalated hypotensive and sedative effects of tizanide, physicians should avoid concomitant administration. Similar effects have been reported also for other fluoroquinolones [59].

### 3.6. Oxazolidinones

Most of the pDDIs involving linezolid and tedizolid are related to pharmacodynamic synergisms (i.e., serotoninergic syndrome with SSRIs, excess of mono-amino oxidase inhibition, etc.) that are beyond the scope of the present review. In the past few years, however, some important PK-driven pDDIs have also emerged. 

#### 3.6.1. Oxazolidinones as Victims of pDDIs

Consistent evidence is now available showing that concomitant administration of rifampicin can significantly reduce the systemic exposure of linezolid, with effect that may persist for up to 2 weeks after rifampicin discontinuation [60,61,62]. In vitro studies documented that rifampicin-inducible drug-metabolizing enzymes have a very minor contribution to linezolid clearance. A large increase in expression of an enzyme (e.g., CYP3A4) that normally plays a relatively modest role in linezolid elimination and/or rifampicin-induced high expression of transport proteins (i.e., p-glycoprotein) have been hypothesized as causes for the observed reduction of linezolid concentrations [60,61,62].

The combination of aztreonam and linezolid in an open-label cross-over study in healthy volunteers resulted in a statistically significant, although probably not clinically relevant, increase of linezolid AUC of approximately 18% [63]. The authors suggest that the mechanism for this interaction is partly explained by a common elimination pathway, i.e., renal excretion. However, the definite mechanism remains unknown.

In 2015 Cojutti and co-workers, by performing univariate analysis, showed that some drug co-treatments were associated with the linezolid trough concentrations, either by lowering the drug exposure (phenobarbital and dexamethasone) or by augmenting it (proton pump inhibitors and amiodarone) [64]. The mechanisms for these pDDIs are poorly understood. As a working hypothesis, it can be speculated that these drugs may affect the expression and/or activity of phase I metabolic enzymes or proteins involved in the distribution of linezolid in the different body compartments. Preliminary evidence is also available from our clinical practice, showing that concomitant administration of meropenem might reduce linezolid systemic exposure (Cattaneo D, Personal Communication).

#### 3.6.2. Oxazolidinones as Perpetrators of pDDIs

The second marketed oxazolidinone tedizolid seems to be associated with a less pDDIs. Indeed, the only clinically relevant pharmacokinetic-driven DDI reported in the drug monograph is the one involving a 55–70% increment in the systemic exposure of rosuvastatin when co-administered with tedizolid [65]. It has been proposed that such DDI is related to the inhibitory effect of tedizolid on the breast cancer resistance protein (BCRP), an efflux carrier involved in the transport of rosuvastatin.

A 15–20% reduction in the exposure of midazolam (CYP3A probe) has been reported in patients concomitantly treated with tedizolid [65]. This pDDI has been, however, considered as of limited clinical relevance.

### 3.7. Aminoglycosides

PDDIs involving aminoglycosides occur mainly at a pharmacodynamic level. Indeed, concomitant administration with nephrotoxic agents (i.e., cisplatin, calcineurin inhibitors, cholinergic agents, loop diuretics, etc.) may worsen renal function and/or cause ototoxicity, whereas concomitant administration with neuromuscular blocking agents or opioids/analgesic may increase the risk of neuromuscular blockade (reviewed in [66]).

Among the oldest aminoglycosides, tobramycin, amikacin, and gentamicin undergo little to no metabolism and, therefore, have a low potential to be victim of pharmacokinetic-driven pDDIs. Some NSAIDs, (such as indomethacin) may, however, increase aminoglycosides plasma concentrations [66].

Plazomicin is a novel semisynthetic parenteral aminoglycoside that inhibits bacterial protein synthesis [67]. It was approved for use in adults with complicated urinary tract infections, including pyelonephritis. Plazomicin is not metabolized by liver microsomes or hepatocytes, has low plasma protein binding (<20%), is extensively renally cleared, with a very low risk of pDDIs resulting from CYP inhibition or induction [66]. In vitro studies showed that plazomicin selectively inhibited multidrug and toxin extrusion (MATE)2-K, MATE1, and OCT2, which are important transporters involved with tubular secretion. However, in a phase I randomized, crossover study in which patients received metformin (which is a probe for these drug transporters and is 90% eliminated via tubular secretion) alone or in combination with plazomicin, no differences on the main metformin pharmacokinetic parameters were observed [68].

### 3.8. Sulfonamides and Trimethoprim

Trimethoprim is a mild inhibitor of OCT2 and of CYP2C8 and sulfamethoxazole is a weak inhibitor of CYP2C9 (reviewed in [69]). Accordingly, concomitant administration of trimethoprim with substrates of OCT2 (i.e., lamivudine, metformine), CYP2C8 (i.e., paclitaxel, amiodarone, dapsone, repaglinide, pioglitazone) and/or CYP2C9 (i.e., warfarin, acenocumarole, phenytoin, glinides) may result in increased overexposure (and eventually increased toxicity). It has been also reported that concomitant administration of trimethoprim increased digoxin exposure (around 20%) by decreasing its renal tubular secretion.

### 3.9. Rifamycins

#### 3.9.1. Rifamycins as Victims of DDIs

The four rifamycins approved for clinical use are rifampicin, rifabutin, rifapentine, and rifaximin (a non-absorbable antibiotic). Rifabutin and rifapentin, available only orally, are of limited relevance in the ICU settings.

Old antiretroviral drugs (amprenavir, indinavir) significantly impacted on the pharmacokinetics of rifabutin and rifampicin (reviewed in [59]), however, these drugs are no longer used. Atazanavir co-administration resulted in an increased rifampicin exposure by 160–250%. Fluconazole had little to no effect on rifampicin pharmacokinetics, however, it increased rifabutin AUC by 82% possibly through inhibition of CYP3A. Posaconazole increased the Cmax and AUC of rifabutin by 31 and 72%, respectively, also possibly through CYP3A4 inhibition.

Co-trimoxazole increased the median AUC of rifampicin by 60%. The bioavailability of rifampicin was reduced by approximately 32% when co-administered with isoniazide. The rifampicin exposure was reduced by co-administration with moxifloxacin, resulting in a decrease of rifampicin AUC by 20%.

#### 3.9.2. Rifamycins as Perpetrators of DDIs

Rifampicin has numerous well-documented clinically significant DDIs associated with its use. Since the initial discovery of several important interactions more than 25 years ago, new interactions continue to be found. Indeed, rifampicin is a potent inducer of CYP enzymes (CYP2B6, 2C8, 2C9, 2C19 and 3A4/) and drug transporters, including P-glycoprotein and OATP1B. Rifampicin may be responsible for strong DDI when co-administered with sensitive CYP substrate drugs and thus increases the hepatic metabolism of several drugs (reviewed in [58]). Moreover, it should be mentioned that rifampicin is also a mild inducer of the uridine diphosphate glucuronosyltransferase (UGT) 1A1 enzymes and interferes with drugs (i.e., integrase inhibitors, mycophenolate, irinotecan, etc.) that are metabolized by this metabolic pathway [70].

Rifampicin is the most potent CYP inducer, and its induction potency is even greater when used at a higher dose of 1200 mg/day, which is common in the therapy of bone and joint infections.

Studies in vitro have consistently documented the activity of rifabutin to induce the expression of metabolic enzymes [59,71]. Based on in vivo findings, rifabutin is considered a less potent inducer than rifampicin and it is likely to cause less clinically-relevant DDIs, but data from comparative studies are limited. Rifabutin at 300 mg/day has lower induction potency than the equivalent dosage of rifampicin (600 mg/day). Consequently, rifabutin is associated with much lower proportions of severe and moderate DDI.

Rifapentine, like other rifamycins, induces the cytochrome P450 system of enzymes—specifically, the CYP3A4, CYP2C8, and CYP2C9 isozymes (reviewed in [72]). It enhances the metabolism and can markedly lower serum concentrations of drugs that are metabolized by these enzymes. One study suggests that the maximal induction of these enzymes occurs within 4 days after receipt of the first dose and returns to the baseline level within 14 days after rifapentine is discontinued [73]; despite limited data, there is no reason to expect rifapentine to induce enzymes faster than rifampin (which requires at least 7 days). Any drug known to have interactions with rifampin should be considered to have similar interactions with rifapentine, unless proven otherwise.

## 4. Tools to Handle pDDIs Involving Antibiotics in ICU

### 4.1. Drug-Interaction Checkers

Several computerized software (some with free access) that detect and rank the severity of DDIs are now available on the web (summarized in Table 3). While these tools are highly desirable, their value depends on how sensitive they are in detecting DDIs and on their accuracy in assessing the type and severity of the interactions. Some of this software has been developed for specific clinical settings, such as those focusing on pDDIs involving antiretroviral, antiviral, and oncology drugs. To the best of our knowledge, no ad hoc checkers have been dedicated to pDDIs involving antimicrobials. Monte-Romea and co-workers recently compared the performance of several drug interaction software platforms to detect and characterize pDDIs involving antimicrobials [74]. They reported a significant variability in the performance of the available platforms in detecting and assessing pDDIs involving antimicrobials, and concluded that, although some checkers have proven to be very accurate, others missed almost half of the explored interactions. Nevertheless, despite these limitations, we strongly suggest always using drug interaction checkers for the routine management of ICU patients on polypharmacy requiring antibiotic treatment.

### 4.2. Therapeutic Drug Monitoring

Therapeutic drug monitoring (TDM) is the clinical practice of measuring drugs in a given biological matrix (usually the blood/plasma/serum) to optimize individual dosage regimens. TDM can indeed provide dosing strategies when a drug is added to or removed from a drug regimen. It can also be useful when an inappropriate combination of drugs has to be continued. Dosing strategies with the use of TDM are related to defined therapeutic ranges that reflect optimal efficacy and safety, or reference ranges that reflect expectations of drug concentrations for a given dose.

Pharmacokinetic-driven pDDIs involving antibiotics as victims may be easily handled in the clinical practice by the TDM of antibiotic plasma concentrations. In fact, therapeutic ranges associated with optimal antimicrobial response and acceptable drug safety have been established in the ICU setting, and several analytical methods are now available on the market to accomplish this task [14,75]. More complicated is the use of TDM for the management of pharmacokinetic-based pDDIs involving antibiotics as perpetrators because this approach requires the availability of analytical methods for the quantification of a heterogeneous set of drugs. TDM can for sure be applied to quantify the potential effects of antibiotics on the systemic disposition of some narrow therapeutic index drugs, such as immunosuppressants, anti-epileptics, some antipsychotics, and other anti-infective agents (i.e., antifungal azoles antiretrovirals), but its use for other drug classes is presently limited.

### 4.3. Physiologically-Based Pharmacokinetic Modelling

Physiologically based pharmacokinetic (PBPK) models represent the body as compartments parameterized based on physiology of tissues and organs, including composition, volumes and blood flows, with the goal to predict the pharmacokinetics of drugs, and allowing simulation of the time course of drug concentrations in plasma and tissues. [76]. Worthy of mention, PBPK is now accepted by regulatory drug agencies (i.e., EMA, FDA) for the prediction of pDDIs. The application of PBPK for the prediction of pDDIs in ICU setting is still in its infancy. However, some preliminary evidence is available showing that this approach can reliably predict, for instance, the pDDIs involving rifampicin as perpetrator or the pharmacokinetics of vancomycin in ICU septic patients [77,78,79].

## 5. Conclusions

Some important pharmacokinetic-driven pDDIs involving antibiotics as victims or perpetrators have been identified. Remarkably, most of them relate to the older antibiotics, whereas novel molecules seem to be associated with a low potential for pDDIs with the exceptions of oritavancin (a potential perpetrator of pDDIs involving narrow therapeutic index drugs metabolized by CYP enzymes) and eravacicline that may be a victim of strong CYP3A inducers (i.e., rifampicin and antifungal azoles).

The accurate prediction of pDDIs can be complex as they may be affected by the concomitant presence of confounding factors, such as patients’ characteristics (i.e., ageing, gender, etc.), drug-induced physiological alterations (i.e., changes in hepatic blood flow, alterations in the protein binding, etc.), ICU-induced pathological alterations (i.e., hypoalbuminemia, augmented renal clearance, etc.), and/or complex dialytic procedures that can significantly impact in the processes of antibiotic distribution, metabolism, and/or elimination, ultimately affecting the clinical relevance of pDDIs, although not specifically in the ICU setting. More recently, some effects of the ethnicity/genetic background, as well as of COVID-19, on drug pharmacokinetics and pDDIs have been also documented [80,81].

A rational approach to the management of these pDDIs might be represented by the application of a “fast-track” clinical pharmacology at the bedside, taking advantage from both the availability of dedicated drug interaction software/checkers, and the TDM of anti-infective and non-anti-infective medications when available [14]. Indeed, preliminary but consistent evidence is now available showing that a combination of the evaluation of pDDIs by clinical pharmacy/pharmacology services and the monitoring of ICU patients is an effective strategy that can be used to optimize drug treatment in this clinical setting [82,83].

## Figures and Tables

**Figure 1 antibiotics-11-01410-f001:**
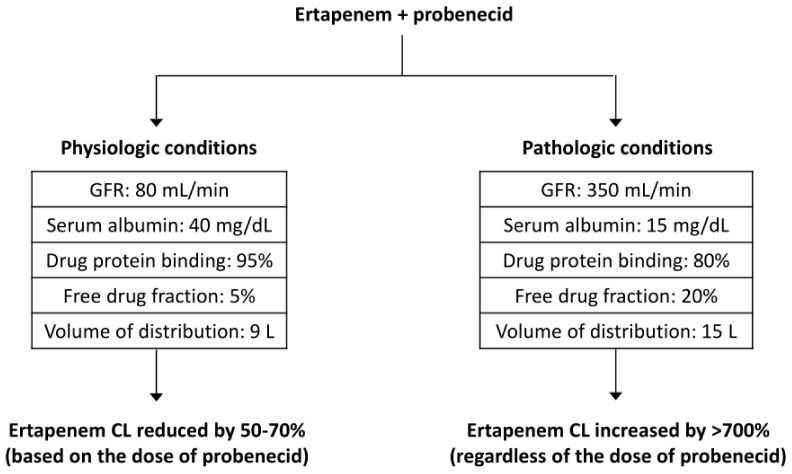
In physiologic conditions (**left side**), concomitant administration of probenecid resulted in a dose-dependent reduction (50–70%) in the clearance (CL) of ertapenem. In pathologic conditions (**right side**), the presence of a severe hypoalbuminemia resulted in a 400% increase in the ertapenem free fraction available for renal excretion. This effect was amplified by the presence of augmented renal clearance (400% increase in the glomerular filtration rate, GFR). Moreover, the patient also gained 6 L of fluids, further diluting the concentrations of ertapenem (the volume of distribution increased by 40%). The net result is a severe reduction in the CL of ertapenem related to the pathological conditions associated with ICU which greatly outweigh the effect of the pDDI related to concomitant probenecid administration.

**Table 1 antibiotics-11-01410-t001:** Potential drug-drug interactions involving novel antibiotics.

Antibiotic	Inhibitory/Inducing Effects	Victim of pDDIs	Perpetrator of pDDIs	Comments
Ceftobiprol	Inhibitor of OATP1B1 and OATP1B3	None	May increase the disposition of OATP substrates	Clinical relevance not demonstrated
Ceftaroline	None	None	None	None
Cefiderocol	Inhibitor of OAT1, OAT3, OCT1, OCT2, MATE-2K, OATP1B3	None	May increase the disposition of substrates of drug transporters	Clinical relevance not demonstrated
Ceftolozane	None	None	None	None
Dalbavancin	None	None	None	The effects of inhibitors on drug transporters have not been studied
Oritavancin	Weak inhibitor of CYP2C9 and CYP2C19, inducer of CYP3A4 and CYP2D6	None	May increase CYP2C9/2C19 substrates and reduce CYP3A4/2D6 substrates	Administer with caution with NTI drugs metabolized by these enzymes
Telavancin	None	None	None	The effects of inhibitors on drug transporters have not been studied
Plazomicin	Inhibitor of MATE2-K, MATE1, OCT2	None	May increase the disposition of substrates of drug transporters	Clinical relevance not demonstrated
Eravacycline	None	Drug exposure reduced by strong CYP3A inducers	None	Increase drug dose of (i.e., 1.5 mg/kg bid) when given with a strong CYP3A inducer
Tedizolid	Inhibition of BCRP	None	May increase the disposition of BCRP substrates	Clinical relevance not demonstrated

pDDIs: potential drug-drug interactions; OATP: organic anion transporting polypeptides; OAT: organic anion transporter; MATE: multidrug and toxin extrusion; OCT: organic cation transporter; CYP: cytochrome P450; BCRP: breast cancer resistance protein; NTI: narrow therapeutic index.

**Table 3 antibiotics-11-01410-t003:** Some of the free web databases that can be used to verify potential drug-drug interactions (pDDIs).

Link	Notes
https://clinicalweb.marionegri.it/intercheckweb	A database that evaluates prescriptive appropriateness in the elderly by considering various aspects of geriatric pharmacology
https://reference.medscape.com/drug-interactionchecker	A “generalist” database that also includes over-the-counter products, some phytotherapeutic agents and supplements
https://www.hiv-druginteractions.org	A database verifying interactions between antiretroviral agents (HIV), and between antiretroviral and non-antiretroviral agents
https://www.hep-druginteractions.org	A database verifying interactions between antiviral agents (HCV), and between antiviral and non-antiviral agents
http://www.drugs.com/drug_interactions.html	A “generalist” database
https://cancer-druginteractions.org/checker	A database verifying interactions between antitumor agents, and between antitumor and non-antitumor agents
http://healthlibrary.uchospitals.edu/Library/DrugReference/DrugInteraction/	A “generalist” database
https://www.rxlist.com/drug-interaction-checker.htm	A “generalist” database
https://www.ddi-predictor.org/predictor/ddi	A “generalist” database
https://stahlonline.cambridge.org/drug_interaction.jsf?page=drugDetails	A “generalist” database that particularly focuses on drugs acting on the central nervous system

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
