# Peer review of "The Issue of Pharmacokinetic-Driven Drug-Drug Interactions of Antibiotics: A Narrative Review"

_antibiotics, 2022, doi:10.3390/antibiotics11101410_

Round 1
Reviewer 1 Report
The manuscript is well written and faces an important topic of the medicine. The auhtors collected many information regarding drug drug interactions involving antibiotics and other drugs with a focused view on patients present in intensive care units.
I just identified some typos:
- line 24. Please add a space after "article."
-line 108. Please replace "increased" with "increase".
-line 239. Please replace "return" with "returns".
-line 248. Please replace "decreased" with "decrease".
-line 388. Please replace "LZD" with "linezolid".
Reviewer 2 Report
The Issue of Pharmacokinetic Driven Drug-Drug Interactions of Antibiotics in the Intensive Care Units
The manuscript presents review of pharmacokinetic Drug-drug interactions (DDIs) with special attention to the DDIs with antibiotics in adult patients in intensive care units. Some solutions were presented at the end. The review took into account the published literature during the last two years and references from the reference list of these articles, which were found to be suitable. In general, DDIs are properly described but very often the main line of the review is lost – namely, critically ill patients. It is necessary to see at least 2-3 sentences for every group of antibiotics which was presented about the probability to observe these DDIs in ICUs. Which critical conditions are considered as the most often seen and what is probability to observe these DDIs? This discrepancy eventually can be avoided by changing the title and by better description of the used criteria for inclusion and exclusion of published articles.
The searching criteria were clearly explained. How many articles were reviewed and how many were included/excluded? The exclusion criteria were not well defined. Please add 2-3 sentences after line 61.
Please, re-phrase the sentence on lines 78-80: “For instance, bi- and trivalent cations (calcium, aluminum, magnesium, iron, etc.), may interfere with the absorption of fluoroquinolones or tetracyclines because of their chelating reactions [10].” - DDIs after oral administration of the drugs are insignificant in ICUs patients because ………….. For instance, it is highly unlikely to observe of DDIs between bi- and trivalent cations (calcium, aluminum, magnesium, iron, etc.) and with fluoroquinolones or tetracyclines at the absorption phase of because of their chelating reactions [10] – Please, do not use directly this text but change the place of the sentences.
Please, add reference in line 194.
Line 209: Most likely pharmacologists will read the paper, but please, give an explanation why statins can be involved in the interaction with Ceftobiprole and give a reference. It will be better if this interaction is easily understandable for clinicians who may read the paper.
Line 211: Please, add first an explanation that a lot of cephalosporins probably will not cause DDIs.
Lines 291-298: This information can be given in 2-3 lines because the described DDIs for orally administered tetracyclines is not relevant for ICUs.
Line 358: Please, revise:
“Ciprofloxacin induced also clozapine and its metabolite N-desmethylclozapine serum concentration through CYP1A2” to “Ciprofloxacin increases also clozapine and its metabolite N-desmethylclozapine serum concentration through CYP1A2”
Line 407: Please, revise:
“(BCRP), an inverse carrier involved in the transport of rosuvastatin.” has to be changed to “(BCRP), an efflux carrier involved in the transport of rosuvastatin”
Reviewer 3 Report
The manuscript is interesting. However, some issues should be mentioned:
1. Don't the Authors think there should be a separate chapter on diuretics. They are widely used in ICU.
2. In my opinion the issue of ethinicity should also be mentioned. The pharmacokinetics of the drugs may be different between the groups.
3. I think that the addition of the information concerning COVID-19 and its impact of the pharmacokinetics of the drugs might be interesting.
Round 2
Reviewer 2 Report
The manuscript has been revised according to the suggestions.
It can be accepted for publication.